# Assessment of COVID-19 data reporting in 100+ websites and apps in India

**Varun Vasudevan**[1], **Abeynaya Gnanasekaran**[1], **Bhavik Bansal**[2], **Chandrakant Lahariya**[3], **Giridara Gopal Parameswaran**[4], **James Zou**[5]*

1 Institute for Computational & Mathematical Engineering, Stanford University, Stanford, California, United States of America, 2 All India Institute of Medical Sciences, New Delhi, India, 3 Public Policy and Health Systems Specialist, New Delhi, India, 4 Center for Disease Dynamics, Economics & Policy, New Delhi, India, 5 Department of Biomedical Data Science, Stanford University School of Medicine, Stanford, California, United States of America

☯ These authors contributed equally to this work.

* jamesz@stanford.edu

## Abstract

India is among the top three countries in the world both in COVID-19 case and death counts. With the pandemic far from over, timely, transparent, and accessible reporting of COVID-19 data continues to be critical for India's pandemic efforts. We systematically analyze the quality of reporting of COVID-19 data in over one hundred government platforms (web and mobile) from India. Our analyses reveal a lack of granular data in the reporting of COVID-19 surveillance, vaccination, and vacant bed availability. As of 5 June 2021, age and gender distribution are available for less than 22% of cases and deaths, and comorbidity distribution is available for less than 30% of deaths. Amid rising concerns of undercounting cases and deaths in India, our results highlight a patchy reporting of granular data even among the reported cases and deaths. Furthermore, total vaccination stratified by healthcare workers, frontline workers, and age brackets is reported by only 14 out of India's 36 subnationals (states and union territories). There is no reporting of adverse events following immunization by vaccine and event type. By showing what, where, and how much data is missing, we highlight the need for a more responsible and transparent reporting of granular COVID-19 data in India.

## Introduction

How many people of each gender have died due to COVID-19 in India? What adverse events have been reported following vaccination? Which states are reporting the number of vacant oxygen beds in their hospitals? Such questions have strong public health implications. Is data reporting from the national and subnational governments in India *granular* enough to answer such questions? We answer that in this paper by documenting and analyzing the reporting of surveillance data [1], vaccination monitoring data [2], and bed availability during the second wave of COVID-19, focusing on granular information. Age-gender distribution for cases and deaths, adverse events following immunization stratified by vaccine and event type, and

**Data Availability Statement:** Our curated dataset used in this study are publicly available at https://github.com/varun-vasudevan/CDRS-India/tree/master/study3_june_2021.

1 / 11

**Funding:** James Zou is supported by discretionary funding from Stanford University. The funders had no role in study design, data collection and analysis, decision to publish, or preparation of the manuscript.

**Competing interests:** The authors declare that there are no competing interests.

number of vacant oxygen beds are few examples of such granular information. Reporting granular COVID-19 data is important for the following reasons.

- It enables public health personnel to track the disease spread, vaccination, and adverse events across different sub-populations [2, 3]. It also allows researchers to gain new insights, and to scrutinize the data to understand the rationale behind the policies put forth by the government.

- Granular data is also more transparent and informative for the general public. Governments cannot give personalized health recommendations to each citizen. The government's advice on mask mandate, lockdowns, and vaccination is designed as a standard recommendation for all the citizens. However, health is a personal matter. If a person is more susceptible to COVID-19, then they need data on age, gender, comorbidities, AEFI (adverse events following immunization), etc., to make informed decisions.

- States in India are not isolated independent regions. People move from one state to another for work and leisure. Therefore, state governments should not just collect data and use them internally, but they should also publish the data so that anyone within the country can use them to make informed decisions.

Our assessment of *reporting quality* of surveillance, vaccination, and vacant bed availability data is timely and important for the following reasons.

- Several articles continue to mention the lack of surveillance data from India [3–5]. Therefore, it is necessary to understand and document what, where, and how much data is missing.

- Assessing the reporting of vaccination monitoring data informs how India fares now and the improvements necessary to overcome future vaccine hesitancy challenges.

- The second COVID-19 wave is significantly larger than the first and led to a severe shortage of resources like oxygen beds [6]. Therefore, it is important to know if the surveillance reporting adapted to the worsening pandemic [7, 8] and if there was reporting on the resources that were in shortage.

Recent studies have highlighted that the official reports in India could be undercounting the true number of COVID-19 cases and death, which raises substantial public health challenges [3, 9, 10]. This work focuses on the complementary question of among the data that is reported, whether useful granularity is provided.

## Methods

Between 22 May and 5 June 2021, we assessed digital platforms hosted by the national and subnational (state and union territory) governments for reporting data on COVID-19 surveillance, vaccination monitoring, and bed availability. Here, digital platforms refer to websites, and mobile applications designed for Android / iOS. We first checked the MyGov mobile app and CoWIN dashboard hosted by the Indian national government to report surveillance and vaccination monitoring data [11, 12]. For each subnational, we then checked their government and health department websites/apps and performed a google search. Overall, we assessed more than 100 digital platforms. At least two authors checked each platform independently on different days and arrived at a consensus on what data is being reported. The complete list of digital platforms that we shortlisted and checked have been made publicly available through the dataset released with this paper. See Text A in S1 Text for more details on data curation.

## Surveillance reporting

Vasudevan et al. developed a framework with 45 indicators to evaluate the reporting quality of COVID-19 surveillance data [7]. We use those indicators in the current study. The indicators check for availability, accessibility, granularity, and privacy violations in the reporting of confirmed, deceased, recovered, quarantine, and critical/ICU (intensive care unit) COVID-19 cases. These five categories indicate possible stages that a susceptible individual can go through during the pandemic.

Availability indicators check for total, daily, and historical data; accessibility indicators check for ease of access and reporting in English; granularity indicators check for total data stratified by age, gender, comorbidity, and districts; and privacy indicator checks if privacy is violated by including personally identifiable information in the reporting. During the assessment all indicators except the following two are scored either a 0 or a 1. The privacy indicator is scored a -1 if there is a privacy violation, else it is scored a 1. The "stratified by comorbidities indicator for deaths" is assigned a score of 1 if binary stratification (presence/ absence of comorbidity) of total deaths is reported. An additional score of 1 is given if more data such as stratification by a list of comorbidities or patient specific comorbidities are reported. We calculate two normalized scores for each subnational as described in [7]. One, a surveillance reporting score, which is the ratio of the total score earned by the subnational from all indicators and the maximum score possible from all indicators. Two, a *granular* surveillance reporting score, which is the ratio of the total score earned by the subnational from granular indicators and the maximum score possible from granular indicators. Both scores range between 0 (low) and 1 (high). During the calculation, the denominator is adjusted if any indicator does not apply to the subnational. For example, stratified by districts does not apply to Chandigarh because it does not have districts. Note that in [7], the surveillance reporting score is referred to as COVID-19 data reporting score and *granular* surveillance reporting score is referred to as granularity score.

To get a handle on the scale of missing granular data, we narrow our focus on the reporting of age and gender for confirmed cases; and age, gender, and comorbidity for deaths. Among subnationals reporting these items, some disaggregate the cumulative numbers by the items; some disaggregate the daily numbers by the items, and the remaining report the items for each individual. Considering all subnationals that report one or more of age, gender, and comorbidity, in any of the three forms mentioned above, we calculate the percentage of cases and deaths for which age, gender, and comorbidity distribution is available as of 5 June, 2021.

## Vaccination reporting

Disaggregated monitoring of vaccination is essential to measure the progress and effectiveness of India's vaccination campaign [2]. We developed a minimal set of indicators to assess the reporting quality of vaccination monitoring data. The indicators reflect recommendations from WHO and LANCET COVID-19 Commission India Task Force [2, 13], and the vaccine operational guidelines from the Ministry of Health and Family Welfare (MoHFW) of India [14].

Indicators are grouped into three dimensions and are as follows. *Availability*: Daily and total vaccination. *Accessibility*: Daily vaccination trend graphic. *Granularity*: 1) Total vaccination stratified by districts and eligibility category (health care workers, front line workers, age 45+, age 18–44). 2) Total AEFI (adverse events following immunization) stratified by vaccine type (Covishield, Covaxin, Sputnik V); and event type (severe, serious).

For all indicators, except the AEFI ones, we check if data is reported separately for each dose (first and second). MoHFW classifies AEFI into three types: *minor* (e.g., pain and swelling

at the injection site, fever), *severe* (e.g., non-hospitalized cases of anaphylaxis, sepsis), and *serious* (e.g., deaths, hospitalizations) [14]. We check for reporting on severe and serious events. Trend graphics are used as an indicator because they are concise and make it easier to identify patterns. Eligibility category refers to the order of eligibility in which vaccines were rolled out in India.

### Vacant bed availability reporting

When resources such as oxygen beds are in shortage [6], it is important to report their vacancy to reduce panic among people in need. Therefore, we checked if subnationals report the number of vacant ICU/oxygen/ventilator beds disaggregated by districts/hospitals.

## Results

### Surveillance reporting

The geographical variation in surveillance reporting scores is shown in Fig 1A. See Table A in S1 Text for each subnational's score. The five number summary of the surveillance reporting score is, minimum = 0.33, first quartile = 0.39, median = 0.46, third quartile = 0.49, and maximum = 0.61. MyGov provides seamless access to total and daily numbers and trend graphics for confirmed, recovered, and deaths for each subnational [11]. However, granular information such as cumulative numbers stratified by districts, age, gender, or comorbidity, is unavailable on MyGov, as summarized in Fig 1B.

Fig 2 lists subnationals in the decreasing order of their *granular* surveillance reporting score. The five number summary of the surveillance reporting score is, minimum = 0, first quartile = 0, median = 0.17, third quartile = 0.22, and maximum = 0.50. Scores from previous assessments are shown for comparison. The northeastern state of Nagaland scored highest by reporting granular data through weekly bulletins. They report cumulative cases and deaths disaggregated by age and gender and cumulative deaths disaggregated by comorbidities, as shown in Fig 3. Nagaland also compares data from 2020 (first wave) with data from 2021 (second wave). In contrast, the lowest scoring subnationals report little or no granular data. As of

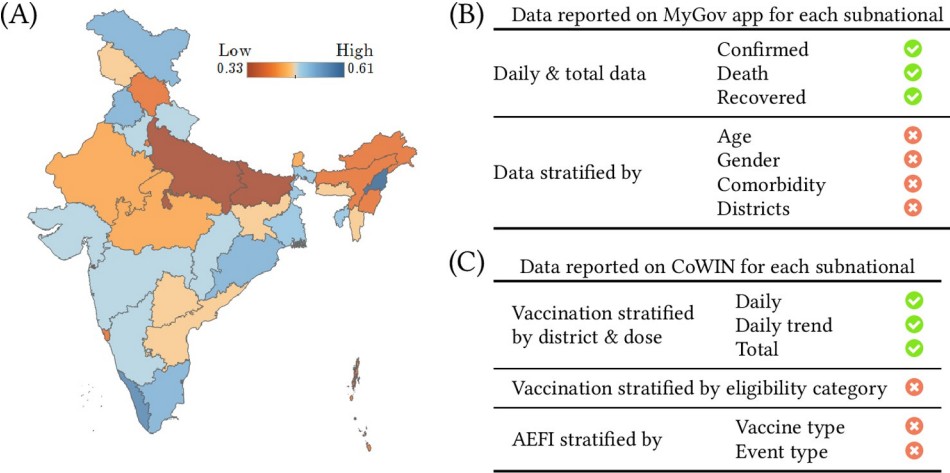

**Fig 1. (A)** Map showing the variation in surveillance reporting score across India. The map was generated using Tableau Desktop software version 2020.2.1 and the boundary information for regions in India was obtained as shapefiles from Datameet (http://projects.datameet.org/maps/). **(B)** Table indicating what surveillance data is being reported (or not) for each subnational on the MyGov app. **(C)** Table indicating the vaccination data reported (or not) for each subnational on the CoWIN dashboard.

| Subnational (State / Union territory) | Granular surveillance reporting score | | | Vaccination monitoring | | | Vacant ICU/oxygen /ventilator bed availability |
|---|---|---|---|---|---|---|---|
| | May 2020 | July 2020 | **June 2021** | By eligibility category | AEFI by vaccine type | AEFI by event type | |
| Nagaland | – | 0.17 | 0.50 ↑ | ✓ | ✗ | ✗ | ✗ |
| Haryana | 0.17 | 0.33 ↑ | 0.33 — | ✗ | ✗ | ✗ | ✓ |
| Tamil Nadu | 0.33 | 0.33 — | 0.33 — | ✗ | ✗ | ✗ | ✓ |
| Kerala | 0.22 | 0.33 ↑ | 0.28 ↓ | ✓ | ✗ | ✗ | ✓ |
| Odisha | 0.28 | 0.28 — | 0.28 — | ✓ | ✗ | ✗ | ✗ |
| Karnataka | 0.39 | 0.50 ↑ | 0.22 ↓ | ✓ | ✗ | ✓ | ✓ |
| Ladakh | 0.21 | 0.22 ↑ | 0.22 — | ✓ | ✗ | ✗ | ✗ |
| Puducherry | 0.22 | 0.22 — | 0.22 — | ✓ | ✗ | ✗ | ✓ |
| Tripura | 0.29 | 0.22 ↓ | 0.22 — | ✗ | ✗ | ✗ | ✗ |
| Uttarakhand | 0.17 | 0.22 ↑ | 0.22 — | ✓ | ✗ | ✗ | ✓ |
| West Bengal | 0.17 | 0.22 ↑ | 0.22 — | ✗ | ✗ | ✗ | ✓ |
| Andhra Pradesh | 0.00 | 0.17 ↑ | 0.17 — | ✗ | ✗ | ✗ | ✓ |
| Chhattisgarh | 0.17 | 0.22 ↑ | 0.17 ↓ | ✗ | ✗ | ✗ | ✓ |
| Gujarat | 0.17 | 0.17 — | 0.17 — | ✓ | ✗ | ✗ | ✗ |
| Jammu & Kashmir | 0.17 | 0.17 — | 0.17 — | ✗ | ✗ | ✗ | ✗ |
| Jharkhand | 0.50 | 0.17 ↓ | 0.17 — | ✓ | ✗ | ✗ | ✗ |
| Madhya Pradesh | 0.17 | 0.17 — | 0.17 — | ✓ | ✗ | ✗ | ✓ |
| Maharashtra | 0.17 | 0.17 — | 0.17 — | ✗ | ✗ | ✗ | ✗ |
| Meghalaya | 0.00 | 0.00 — | 0.17 ↑ | ✗ | ✗ | ✗ | ✗ |
| Mizoram | – | 0.07 | 0.17 ↑ | ✓ | ✗ | ✗ | ✗ |
| Punjab | 0.17 | 0.17 — | 0.17 — | ✓ | ✗ | ✗ | ✓ |
| Telangana | 0.00 | 0.00 — | 0.17 ↑ | ✗ | ✗ | ✗ | ✓ |
| Rajasthan | 0.11 | 0.11 — | 0.11 — | ✗ | ✗ | ✗ | ✓ |
| Sikkim | – | 0.00 | 0.11 ↑ | ✗ | ✗ | ✗ | ✗ |
| Andaman & Nicobar | 0.00 | 0.00 — | 0.00 — | ✗ | ✗ | ✗ | ✗ |
| Arunachal Pradesh | – | 0.00 | 0.00 — | ✗ | ✗ | ✗ | ✗ |
| Assam | 0.17 | 0.17 — | 0.00 ↓ | ✗ | ✗ | ✗ | ✗ |
| Bihar | 0.00 | 0.00 — | 0.00 — | ✗ | ✗ | ✗ | ✓ |
| Chandigarh | 0.00 | 0.00 — | 0.00 — | ✗ | ✗ | ✗ | ✓ |
| Delhi | 0.00 | 0.00 — | 0.00 — | ✗ | ✗ | ✗ | ✓ |
| DNH & DD | – | 0.22 | 0.00 ↓ | ✗ | ✗ | ✗ | ✓ |
| Goa | 0.00 | 0.06 ↑ | 0.00 ↓ | ✗ | ✗ | ✗ | ✓ |
| Himachal Pradesh | 0.00 | 0.00 — | 0.00 — | ✓ | ✗ | ✗ | ✓ |
| Lakshadweep | – | – | 0.00 | ✗ | ✗ | ✗ | ✗ |
| Manipur | – | 0.00 | 0.00 — | ✓ | ✗ | ✗ | ✗ |
| Uttar Pradesh | 0.00 | 0.00 — | 0.00 — | ✗ | ✗ | ✗ | ✗ |

**Symbols:** – for score unavailable; — for no change; ↑ for increase; ↓ for decrease; ✓ for *is reporting*; and ✗ for *is not reporting*

**Fig 2. Subnationals sorted in the decreasing order of granular surveillance reporting score from the current assessment (June 2021).** The scores from previous assessments (2020) are shown for comparison [7, 8]. The table also shows which subnationals are reporting (or not) vaccination coverage stratified by eligibility category; AEFI stratified by vaccine and event type; and vacant ICU/oxygen/ventilator bed availability disaggregated by districts/hospitals.

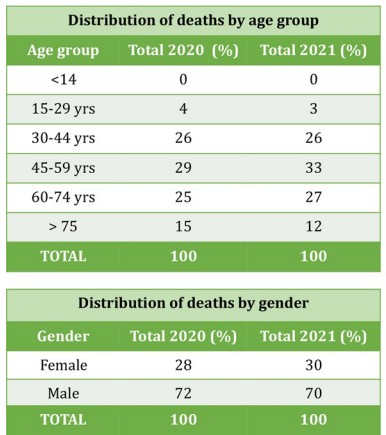

**Distribution of deaths by age group**

| Age group | Total 2020 (%) | Total 2021 (%) |
|---|---|---|
| <14 | 0 | 0 |
| 15-29 yrs | 4 | 3 |
| 30-44 yrs | 26 | 26 |
| 45-59 yrs | 29 | 33 |
| 60-74 yrs | 25 | 27 |
| > 75 | 15 | 12 |
| TOTAL | 100 | 100 |

**Distribution of deaths by gender**

| Gender | Total 2020 (%) | Total 2021 (%) |
|---|---|---|
| Female | 28 | 30 |
| Male | 72 | 70 |
| TOTAL | 100 | 100 |

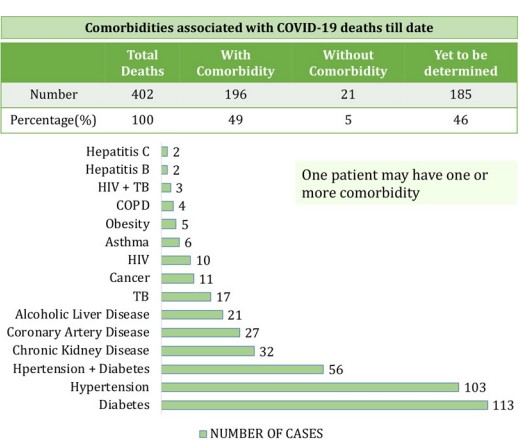

**Comorbidities associated with COVID-19 deaths till date**

| | Total Deaths | With Comorbidity | Without Comorbidity | Yet to be determined |
|---|---|---|---|---|
| Number | 402 | 196 | 21 | 185 |
| Percentage(%) | 100 | 49 | 5 | 46 |

One patient may have one or more comorbidity

- Hepatitis C: 2
- Hepatitis B: 2
- HIV + TB: 3
- COPD: 4
- Obesity: 5
- Asthma: 6
- HIV: 10
- Cancer: 11
- TB: 17
- Alcoholic Liver Disease: 21
- Coronary Artery Disease: 27
- Chronic Kidney Disease: 32
- Hpertension + Diabetes: 56
- Hypertension: 103
- Diabetes: 113

NUMBER OF CASES

**Fig 3. Age, gender, and comorbidity data for deaths provided in the weekly bulletin of Nagaland government on 5 June 2021 as examples of high-quality granular surveillance reporting.**

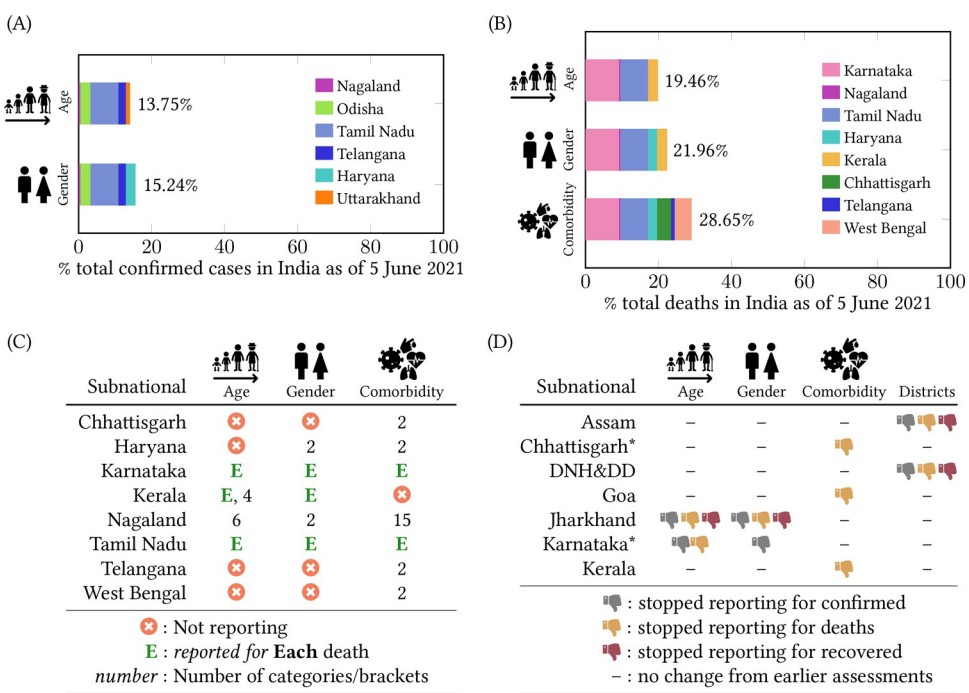

**Fig 4.** **(A)** Shows the % of cases for which age and gender distribution are available. Each subnational reporting that data is represented by a colored rectangle whose width denotes the % of total cases in India reported in that subnational. **(B)** Shows the % of deaths for which age, gender, and comorbidity data distribution are available. Each subnational reporting that data is represented by a colored rectangle whose width denotes the % of total deaths in India reported in that subnational. **(C)** Table showing the variation in the format of reporting of age, gender, and comorbidity among the subnationals that are reporting those data for the deaths. **(D)** Subnationals that stopped reporting total data stratified by age, gender, comorbidity or district after either of the surveillance reporting assessments conducted in 2020 [7, 8]. *See Text D in S1 Text for more details.

5 June 2021, age and gender distribution are available for less than 22% of cases and deaths, and comorbidity distribution is available for less than 30% of deaths. Subnationals that report both age and gender distribution for cases are: Nagaland, Odisha, Tamil Nadu, and Telangana. Similarly, subnationals that report both age and gender distribution for deaths are: Karnataka, Nagaland, Tamil Nadu, and Kerala. See Fig 4A–4D for a compact summary of granular data availability and Text C in S1 Text for additional details. Even now, some subnationals do not report data stratified by districts. The tabular dataset released with this paper provides a comprehensive summary of what data each subnational is reporting.

## Privacy violations

Chandigarh and Haryana are violating privacy by including individually identifiable information in their reporting. Chandigarh continues to release a document (http://chandigarh.gov. in/health_covid19.htm) containing the name and address of people who have completed/ under quarantine. The document is over a thousand pages with over 100,000 entries. Haryana is releasing a document (https://haraadesh.nic.in/) containing the name, age, gender, and address of cases from the Jhajjar district (see Fig A in S1 Text).

## Vaccination reporting

CoWIN, launched in 2021, is a cloud-based information technology solution for planning, implementing, monitoring, and evaluating COVID-19 vaccination [14]. CoWIN dashboard

reports the following for each subnational, district, and dose. Daily and total vaccination numbers, and daily vaccination trend graphics. CoWIN does not report total vaccination stratified by eligibility category for each dose. For AEFI, CoWIN reports daily AEFI numbers and the cumulative percentage. The number of severe and serious events disaggregated by vaccine type is missing (Fig 1C). Only 14 out 36 subnationals report on their digital platforms the total vaccination stratified by eligibility category for each dose. They are Nagaland, Kerala, Odisha, Karnataka, Ladakh, Puducherry, Uttarakhand, Gujarat, Jharkhand, Madhya Pradesh, Mizoram, Punjab, Himachal Pradesh, and Manipur. Karnataka is the only subnational that is reporting the number of severe and serious AEFI cases. AEFI reporting stratified by vaccine type is absent on all subnational platforms. These findings are summarized in Fig 2.

## Vacant bed availability reporting

20 out of 36 subnationals report vacant bed availability by hospitals and frequently update them. They are Haryana, Tamil Nadu, Kerala, Karnataka, Puducherry, Uttarakhand, West Bengal, Andhra Pradesh, Chhattisgarh, Gujarat, Madhya Pradesh, Punjab, Telangana, Rajasthan, Bihar, Chandigarh, Delhi, Goa, Himachal Pradesh, Dadra and Nagar Haveli and Daman and Diu. These results are also summarized in Fig 2. It is a commendable effort from these subnationals to ensure the effective utilization of resources. Other subnationals are either not publishing any data on vacant bed availability or are reporting the total/vacant number of beds without classifying them. We encourage these subnationals to be more granular in reporting.

## Discussion

This is the largest study of its kind to assess the quality of COVID-19 data reporting in India. We did a comprehensive assessment of 100+ national and subnational government digital platforms (web and mobile) to identify what is present and what is missing in the reporting of surveillance data, bed availability, and vaccination monitoring data.

Overall, the quality of surveillance reporting has improved since 2020. Median surveillance reporting score has increased from 0.26 in May 2020 [7] and 0.30 in July 2020 [8]. This increase is primarily due to the consistent availability of high-level surveillance data through MyGov. However, the reporting of granular information such as age, gender, and comorbidity continues to be poor.

Age and gender distribution is available for about 1 in 5 cases and deaths in India. Similarly, comorbidity distribution is available for about 1 in 3 deaths. That is a staggeringly low number for a country with more than 344 thousand deaths. Essentially, we do not know even the basic information about who is getting infected and who is dying. This limitation has important implications.

1. It prohibits researchers from tracking age-gender specific trends, identifying high-risk subgroups, and validating hypotheses on infection fatality rates [3, 5].

2. It is difficult to understand the effect of the virus on the age group below 18 without data on the age distribution of cases, deaths, and ICU cases. This is important as schools are reopening in India.

3. Number of new confirmed cases per 100,000 population per week and number of COVID-19-attributed deaths per 100,000 population per week are two primary indicators to assess the level of community transmission as per WHO. It is important to track these indicators at the district level because subnationals in India are big and health care facilities vary significantly within a subnational. Therefore, states should publish data at least at the district level.

Going further, disaggregating cases and deaths by vaccination status (fully/partially/not vaccinated) is also essential to estimate the vaccine effectiveness in various sub-groups. Maharashtra, the state with the most deaths, does not report the age, gender, and comorbidity distribution. Even among subnationals reporting granular data for deaths, differences in the format of reporting make it difficult for comparison. See Fig 4C for a visual summary of the differences in the format of reporting.

A few subnationals have discontinued reporting certain granular items since the assessments in 2020 (Fig 4D). We highlight three specific instances. First, Karnataka, the state with the best surveillance reporting in 2020 [7, 8], is no longer publishing war-room bulletins that had age and gender data for cases. Second, Kerala has stopped reporting comorbidity for deaths. There are claims that Kerala is undercounting deaths by attributing a portion of them as death due to comorbidity [15, 16]. The removal of comorbidity data could strengthen such claims. Third, Jharkhand, a model state for granular reporting in the initial months, stopped reporting age and gender data as the pandemic worsened. It is important to scrutinize these changes in reporting to understand the bottlenecks or motives that led to the changes.

On the one side, there is inadequate reporting of essential granular data like age and gender distribution. On the other side, personally identifiable information is being published by subnationals like Chandigarh and Haryana. The public health benefits of the personally identifiable information released by these subnationals are unclear. Data reported by the government should include only the information necessary for public health activities [17]. Reporting personal data can discourage people from cooperating with the government or lead to discrimination against specific people [18]. For example, it might be possible to infer religion from names of some people and target specific groups leading to communal violence. India has already seen communal violence in the context of COVID-19 [19].

The quality of surveillance reporting in India has been analyzed extensively in three studies, including the current one. The first two studies were during the first wave of COVID-19 (roughly 3 and 6 months into the pandemic) [7, 8], and the current study was during the second wave (after 15 months). Two crucial lessons that we learned collectively from these assessments are as follows. First, subnational governments are unlikely to make much progress in granular surveillance reporting without an official guideline from the central government on what data they have to report publicly. In fact, without someone to hold the subnational governments accountable, they can even switch from good to poor reporting practices (e.g., Karnataka and Jharkhand). Second, while official documents from the government, including a recent white paper from NITI Aayog (Vision 2035: Public Health Surveillance in India) [20], embrace the importance of privacy, there is an evident lack of awareness about privacy among officials releasing surveillance data.

We make three comments about the reporting of vaccination monitoring data. First, through the CoWIN dashboard, anyone can access vaccination coverage data for all subnationals and districts. It is a remarkable feat for such a large country. Second, governments should at least report vaccination coverage disaggregated by eligibility category. In the coming months, more disaggregated reporting based on gender, pre-existing conditions (comorbidities, pregnancy), socioeconomic, rural-urban, and other equity factors are necessary to ensure no sub-groups are left behind [2]. Third, there is an urgent need for reporting AEFI by vaccine type, sub-population affected, gender, and severity. Detailed and transparent AEFI data can increase citizens' confidence in vaccines, especially as these vaccines are still in the emergency use authorization phase [21]. A large part of the success of polio elimination in India can be credited to disaggregated program data and a robust AEFI reporting system.

A similar study performed by Rocco et al. evaluated the quality of COVID-19 surveillance data across 15 federal democracies, including India [22]. Their evaluation using 13 indicators

found a statistically significant association between subnational data quality and critical public health system capacity indicators. Countries such as the United States, Canada, Belgium, and Germany that are known to have substantial public health capacity and infrastructure scored higher on data quality. In contrast, countries like Argentina, India, and Malaysia scored significantly below the median score.

Our study has several limitations that would be interesting to address in follow up research. One main analysis limitation is that we can not evaluate the effect of data reporting quality on the containment of the virus. Therefore, our results should not be interpreted as "good reporting means good control of the pandemic." While transparent and timely reporting of data is necessary, it is not sufficient. As discussed in the introduction, there could be a substantial number of COVID-19 cases and deaths that are under-reported, and we do not quantify these in our analysis. Our assessment is restricted to national and subnational (state and union territory) platforms and does not include district platforms.

## Conclusions

By not reporting granular details, governments are making a choice to make certain information invisible to the scientific community and the public. One interesting direction for future research is to explore what decisions shape how governments in India are reporting COVID-19 surveillance and vaccination monitoring data [23, 24].

As researchers and health professionals, our goal here is to advocate for change through measurement. Through a semi-quantitative approach, we showed the specifics and magnitude of missing granular data across India. Our findings provide the largest and most recent evidence for lack of granularity in India's COVID-19 data reporting. Governments in India should recognize the importance of reporting granular data and make it a priority before the next wave of COVID-19. As a start, we recommend reporting the following. First, age and gender distribution for cases and deaths, and comorbidities for deaths. Second, details of serious/severe AEFI cases. Third, vaccination coverage for each dose stratified by eligibility category.

## Supporting information

**S1 Text. Information on shortlisting digital platforms, surveillance reporting score for each subnational, calculating the amount of missing granular surveillance data, additional notes on Figs 2 and 4, suggestions on granular reporting of testing data, and privacy violation in the reporting from Haryana.**
(PDF)

## Acknowledgments

We want to thank healthcare workers across the globe for their efforts during the pandemic. We also thank the members of India COVID SOS and the Stanford community for their support and insightful feedback on a version of the draft.

## Author Contributions

**Conceptualization:** Varun Vasudevan, Abeynaya Gnanasekaran, James Zou.

**Data curation:** Varun Vasudevan, Abeynaya Gnanasekaran, Bhavik Bansal.

**Formal analysis:** Varun Vasudevan, Abeynaya Gnanasekaran, James Zou.

**Funding acquisition:** James Zou.

**Investigation:** Varun Vasudevan, Abeynaya Gnanasekaran, James Zou.

**Methodology:** Varun Vasudevan, Abeynaya Gnanasekaran, Bhavik Bansal, Chandrakant Lahariya, Giridara Gopal Parameswaran, James Zou.

**Software:** Varun Vasudevan, Abeynaya Gnanasekaran.

**Supervision:** James Zou.

**Validation:** Varun Vasudevan, Abeynaya Gnanasekaran, Bhavik Bansal, Chandrakant Lahariya, Giridara Gopal Parameswaran, James Zou.

**Visualization:** Varun Vasudevan, Abeynaya Gnanasekaran.

**Writing – original draft:** Varun Vasudevan, Abeynaya Gnanasekaran, Bhavik Bansal, Chandrakant Lahariya, Giridara Gopal Parameswaran.

**Writing – review & editing:** Varun Vasudevan, Abeynaya Gnanasekaran, Bhavik Bansal, Chandrakant Lahariya, Giridara Gopal Parameswaran, James Zou.

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
