## [Decision Letter · Decision Letter 0]

12 Oct 2021

PGPH-D-21-00483

Assessment of COVID-19 data reporting in 100+ websites and apps in India

Dear Dr. Gnanasekaran,

Thank you for submitting your manuscript to PLOS Global Public Health. After careful consideration, we feel that it has merit but does not fully meet PLOS Global Public Health’s publication criteria as it currently stands. Therefore, we invite you to submit a revised version of the manuscript that addresses the points raised during the review process.

One of the reviews highlights careful rewriting/framing of the background, the discussion as well as the presentation of the results needs to be reassessed in line with the review. 

We look forward to receiving your revised manuscript.

Kind regards,

Prashanth Nuggehalli Srinivas, MBBS, MPH, PhD

Academic Editor

Journal Requirements:

1. Please amend your detailed Financial Disclosure statement. This is published with the article, therefore should be completed in full sentences and contain the exact wording you wish to be published.

i). State what role the funders took in the study. If the funders had no role in your study, please state: “The funders had no role in study design, data collection and analysis, decision to publish, or preparation of the manuscript.”

Additional Editor Comments (if provided):

Reviewers' comments:

Reviewer's Responses to Questions

**Comments to the Author**

1. Does this manuscript meet PLOS Global Public Health’s publication criteria? Is the manuscript technically sound, and do the data support the conclusions? The manuscript must describe methodologically and ethically rigorous research with conclusions that are appropriately drawn based on the data presented.

Reviewer #1: Partly

Reviewer #2: Yes

2. Has the statistical analysis been performed appropriately and rigorously?

Reviewer #1: No

Reviewer #2: Yes

3. Have the authors made all data underlying the findings in their manuscript fully available (please refer to the Data Availability Statement at the start of the manuscript PDF file)?

Reviewer #1: No

Reviewer #2: Yes

4. Is the manuscript presented in an intelligible fashion and written in standard English?

Reviewer #1: No

Reviewer #2: Yes

5. Review Comments to the Author

Reviewer #1: The area is interesting but the manuscript does not follow the scientific guidelines of a research paper. It appears more like a book chapter. Authors need to cover the apps in detail and present a framework for new app-makers to follow through their research efforts. Currently only reporting is being done which does not add much value to the existing literature.

Reviewer #2: The authors present an important topic examining the quality of reporting of COVID-19 data (including surveillance, vaccination monitoring, and hospital bed availability) in over 100 government platforms (web and mobile) from India. The article flows well and there are limited typos. While the article shows great potential and the subject matter is important, there are several modifications that are suggested. In the current state, a revision and resubmission is recommended.

Introduction:

Additional information about why this data this important and the potential impact it has for different stakeholder groups would strengthen the Introduction.

Methods:

Additional information about methods and analysis would be helpful

Did one person code all of the websites or multiple people? If multiple people coded the websites, how did you ensure IRR?

Surveillance Reporting – I’m not sure accessibility is the appropriate term or I would recommend better defining “accessibility”. Did you use Web Content Accessibility Guidelines or look at the reading levels of content presented?

Results:

There are so many tables and figures (and figures within figures) that it is difficult to comprehend all the information. Consider removing or combining some tables/figures. Alternatively, consider turning this paper into multiple manuscripts.

Overall, additional text in the Results section would be helpful – rather than pointing to all the tables and figures. Additionally, if the data are normally distributed, please report means and standard deviations or medians and ranges or interquartile ranges for data that are not normally distributed.

Discussion:

The “so what” and implications of this study need to be strengthened. The commentary on the results reads awkwardly and is a bit of a stretch in the current state.

6. PLOS authors have the option to publish the peer review history of their article (what does this mean?). If published, this will include your full peer review and any attached files.

**Do you want your identity to be public for this peer review?** For information about this choice, including consent withdrawal, please see our Privacy Policy.

Reviewer #1: No

Reviewer #2: No

---

## [Decision Letter · Decision Letter 1]

16 Dec 2021

PGPH-D-21-00483R1

Assessment of COVID-19 data reporting in 100+ websites and apps in India

Dear Dr. Gnanasekaran,

Thank you for submitting your manuscript to PLOS Global Public Health. After careful consideration, we feel that it has merit but does not fully meet PLOS Global Public Health’s publication criteria as it currently stands. Therefore, we invite you to submit a revised version of the manuscript that addresses the points raised during the review process.

Please note minor changes requested in the latest reviews. Please carefully go through these before proceeding to publication. 

We look forward to receiving your revised manuscript.

Kind regards,

Prashanth Nuggehalli Srinivas, MBBS, MPH, PhD

Academic Editor

Journal Requirements:

Additional Editor Comments (if provided):

Please address all reviewer inputs, particularly the minor suggestions from reviewer 4 in your final version

Reviewers' comments:

Reviewer's Responses to Questions

**Comments to the Author**

1. If the authors have adequately addressed your comments raised in a previous round of review and you feel that this manuscript is now acceptable for publication, you may indicate that here to bypass the “Comments to the Author” section, enter your conflict of interest statement in the “Confidential to Editor” section, and submit your "Accept" recommendation.

Reviewer #3: (No Response)

Reviewer #4: (No Response)

2. Does this manuscript meet PLOS Global Public Health’s publication criteria? Is the manuscript technically sound, and do the data support the conclusions? The manuscript must describe methodologically and ethically rigorous research with conclusions that are appropriately drawn based on the data presented.

Reviewer #3: Yes

Reviewer #4: Yes

3. Has the statistical analysis been performed appropriately and rigorously?

Reviewer #3: N/A

Reviewer #4: Yes

4. Have the authors made all data underlying the findings in their manuscript fully available (please refer to the Data Availability Statement at the start of the manuscript PDF file)?

Reviewer #3: Yes

Reviewer #4: Yes

5. Is the manuscript presented in an intelligible fashion and written in standard English?

Reviewer #3: Yes

Reviewer #4: Yes

6. Review Comments to the Author

Reviewer #3: This manuscript does a very good job of objectively assessing the quality and granularity of COVID-19 related data released by states and union territories of India. It reads well as a continuation of the previous article in BMC Public Health in which the COVID-19 Data Reporting Score was developed. In addition to focusing on granularity of surveillance data, the present manuscript also analyzes vaccine related data and bed vacancy data. "Lack of data" is a widely heard complaint and the figures included in this manuscript illustrate what that really means.

While systematically and painstakingly scoring and analyzing each subnation on one side, the authors have also highlighted the best and the worst practices with specific examples. The methodology is simple and straightforward although it requires a couple of careful readings of the textual description to realize that. It is possibly an unfortunate constraint of the medium of publication that useful details are spread across multiple locations (text, figure, supplementary file, and data repository).

The "men and women" in the opening sentence seems to reinforce an outdated binary. The color palette in Fig 1A is not very friendly to this reviewer who has red-green color deficiency. It is not stated whether the authors have pointed out the privacy violations to the concerned authorities and given them a chance to rectify.

This manuscript legitimizes in academic terms the rather political demand for open data. And the poise with which the authors have done so makes it an important contribution in the field of open data for health governance and public policy.

Reviewer #4: Overall comments: The authors have used a novel and reproducible approach to demonstrate how the COVID-19 data is reported. The topic is relevant to the current context and informs readers about how the data is being used. Since the authors have accessed various websites for shortlisting digital platforms, it would be required for them to mention the exact time and date that these websites were accessed, as websites evolve over a period of time (dynamic in nature).

Methods: Written in a detailed and reproducible manner.

Line 81: It would be good to provide a definition for web-based and mobile based digital platforms along with few examples as most platforms these days are web-based, but accessible through mobile. I am sharing a link that may be referred to https://careerfoundry.com/en/blog/web-development/what-is-the-difference-between-a-mobile-app-and-a-web-app/

Discussion: In the methods section the authors mention that Vasudevan et al., developed a framework of 45 indicators to evaluate the reporting quality of COVID-19 surveillance data. Did the authors explore if the framework has been used for settings other than India? The authors may bring the findings of those studies (if any) in the discussion section. It would be imperative to cite studies from other regions on this topic (if any). If there are not many studies, then point towards the paucity in literature and that adds value to your research.

References:

The reference 6 and 18 need attention. "The Lancet" should not be written as "Lancet T".

7. PLOS authors have the option to publish the peer review history of their article (what does this mean?). If published, this will include your full peer review and any attached files.

**Do you want your identity to be public for this peer review?** For information about this choice, including consent withdrawal, please see our Privacy Policy.

Reviewer #3: **Yes: **Akshay S Dinesh

Reviewer #4: No

---

## [Editor Report · Decision Letter 2]

14 Mar 2022

Assessment of COVID-19 data reporting in 100+ websites and apps in India

PGPH-D-21-00483R2

Dear Ms. Gnanasekaran,

We are pleased to inform you that your manuscript 'Assessment of COVID-19 data reporting in 100+ websites and apps in India' has been provisionally accepted for publication in PLOS Global Public Health.

Best regards,

Prashanth Nuggehalli Srinivas, MBBS, MPH, PhD

Academic Editor